# Is the HPV-test more cost-effective than cytology in cervical cancer screening? An economic analysis from a middle-income country

**Diama Bhadra Vale**[1]*, **Marcus Tolentino Silva**[2], **Michelle Garcia Discacciati**[1], **Ilana Polegatto**[1], **Julio Cesar Teixeira**[1], **Luiz Carlos Zeferino**[1]

**1** Obstetrics and Gynecology Department, State University of Campinas, Campinas, Brazil, **2** Program in Pharmaceutical Science, University of Sorocaba, Sorocaba, Brazil

* dvale@unicamp.br

## Abstract

### Objective

To report a modelling study using local health care costs and epidemiological inputs from a population-based program to access the cost-effectiveness of adopting hrHPV test.

### Methods

A cost-effectiveness analysis based on a microsimulation dynamic Markov model. Data and costs were based on data from the local setting and literature review. The setting was Indaiatuba, Brazil, that has adopted the hrHPV test in place of cytology since 2017. After calibrating the model, one million women were simulated in hypothetical cohorts. Three strategies were tested: cytology to women aged 25 to 64 every three years; hrHPV test to women 25–64 every five years; cytology to women 25–29 years every three years and hrHPV test to women 30–64 every five years (hybrid strategy). Outcomes were Quality-adjusted life-years (QALY) and Incremental Cost-Effectiveness Ratio (ICER).

### Results

The hrHPV testing and the hybrid strategy were the dominant strategies. Costs were lower and provided a more effective option at a negative incremental ratio of US$ 37.87 for the hybrid strategy, and negative US$ 6.16 for the HPV strategy per QALY gained. Reduction on treatment costs would influence a decrease in ICER, and an increase in the costs of the hrHPV test would increase ICER.

### Conclusions

Using population-based data, the switch from cytology to hrHPV testing in the cervical cancer screening program of Indaiatuba is less costly and cost-effective than the old cytology program.

**Data Availability Statement:** The data underlying this study are available on Dryad (www.doi.org/10.17605/osf.io/f3ygd).

**Funding:** This project is funded by UNICAMP (Women's Hospital), Indaiatuba City (SUS), and Roche Diagnostics, as detailed below: This study proposal (screening program implementation and cost-effectiveness analyses) was designed by researchers from UNICAMP and introduced by the Indaiatuba City Hall. Both UNICAMP and the municipality use the existing and functioning structure to place the new screening program and carry out the proposed study at no additional cost. The supplies and equipment required to perform HPV testing (for one round in five years), computer system development support, two lab technicians, and a screening program coordinator are provided or supported by Roche Diagnostics. Roche Diagnostics supports an external statistician defined by researchers to develop a model and perform the planned cost-effectiveness analyses. There are no planned compensations or cash transfers provided to any institution or researchers stated in the parties' cooperation agreement.

**Competing interests:** This project is funded by UNICAMP (Women's Hospital), Indaiatuba City (SUS), and Roche Diagnostics®, as detailed below: This study proposal (screening program implementation and cost-effectiveness analyses) was designed by researchers from UNICAMP and introduced by the Indaiatuba City Hall. Both UNICAMP and the municipality use the existing and functioning structure to place the new screening program and carry out the proposed study at no additional cost. The supplies and equipment required to perform HPV testing (for one round in five years), computer system development support, two lab technicians, and a screening program coordinator are provided or supported by Roche Diagnostics®. Roche Diagnostics® supports an external statistician defined by researchers to develop a model and perform the planned cost-effectiveness analyses. There are no planned compensations or cash transfers provided to any institution or researchers stated in the parties' cooperation agreement. Roche Diagnostics® support does not alter our adherence to PLOS ONE policies on sharing data and materials.

# Introduction

Screening has reduced cervical cancer incidence rates in the past 50 years [1]. Although many screening activities have been taking place in low and middle-income countries (LMIC), this reduction was not as prominent as those observed in high-income countries (HIC), where population-based organized screening programs were adopted [2]. The fragility of the public health systems in LMIC is a barrier to high-quality programs [3]. The adoption of more efficient technologies can improve screening performance, avoiding abrupt changes in the current practices.

The choice of the test to be offered is a core constituent of a screening program. It must be sensitive and present a high negative predictive value. It should be easy to perform and widely accepted [4]. Cytology is the conventional test in cervical cancer screening, but its sensitivity is low [5]. The recognition of the fundamental role of the HPV infection in the natural history of cervical cancer supports the use of high-risk DNA-HPV (hrHPV) tests in screening. Randomized clinical trials have shown that, compared to cytology, the use of hrHPV-test results in higher rates of detection of precursor lesions, higher negative predictive value, and decreased cervical cancer mortality rates [6, 7].

The World Health Organization (WHO) recommends that, when resources are available, settings using cytology in screening should consider the transition to hrHPV testing based programs [8]. HIC cost-effectiveness studies have demonstrated the superiority of using hrHPV testing compared to cytology [9–16]. In low-income countries, health systems are usually developing, whereas, in middle-income countries (MIC), a better health framework makes it possible to design and implement more sophisticated programs. Unfortunately, population-based cost-effectiveness studies are scarce in MIC, limiting the incorporation of new technologies [3, 17].

Indaiatuba is a city far 100 km from São Paulo's city and represents a medium-sized Brazilian city. In October 2017, the public health system adopted to all women targeted primary hrHPV testing in the screening program, replacing the cytology-based program [18]. The following research question was designed: which test is more cost-effective in cervical cancer screening at the public health system: cytology or hrHPV test? A modelling study using local health care costs and epidemiological inputs from the program was developed to answer the question. The results can support the planning for regional strategies and intend to reference others, optimizing resource allocation.

# Material and methods

## Context and population

The present study is a cost-effectiveness analysis based on a microsimulation dynamic Markov model. Data and costs were based on data from the local setting and literature review. Indaiatuba city is located in the state of São Paulo, Brazil, and has about 250,000 inhabitants. The main economic activities are industry and commerce, and the Human Development Index is 0.79, above the average of the state and the country [19].

In Brazil, the public health system runs through the 'Unified Health System' (SUS), free of charge to all citizens. It is hierarchical, with a gateway through primary care, where cancer screening activities take place. The population can also co-use private health services through health insurance paid by the customer or employer.

In 2017 Indaiatuba adopted the hrHPV test to replace cytology in cervical cancer screening. A detailed description of the program was previously published by Teixeira and colleagues [18]. In summary, the hrHPV test chosen provides individual results on the highest risk

genotypes—HPV 16 and HPV 18—and an aggregated results on the twelve other hr-HPV genotypes (types 31, 33, 35, 39, 45, 51, 52, 56, 58, 59, 66 and 68). The negative test indicates a repetition in five years. A specific law regulated the new program. Since SUS has not yet incorporated the hrHPV test, the municipality was responsible for financing the test, subsidized by the Roche® laboratory. Parallel to the implementation, a research team from the University of Campinas assessed the program's performance and cost-effectiveness [18].

In Brazil, the target women for cervical cancer screening are those aged 25 to 64 [20]. The hrHPV test performance is lower in women aged 25 to 29 due to the reduction of its specificity and high prevalence of transient HPV infections. Thus, many recommend that the test be offered only from 30 years old, and cytology should to women aged 25 to 29 [8, 21, 22]. However, the policymakers of Indaiatuba considered that if both technologies were available (hrHPV and cytology), it would induce the misuse of them. Furthermore, it would be necessary to maintain both logistics at the same time. They choose a more straightforward design for the program: the hrHPV test offered to all women aged 25 to 64, and cytology only for triage of positive results (reflex test) [18].

## Study perspective

The incremental cost-effectiveness ratio (ICER) was assessed from the financing system's perspective, the Brazilian "SUS". It included direct costs such as screening logistics and referral for treatment: outpatient care, treatment procedures, relapse treatment, complications procedures, inpatient care and medications. Indirect and intangible costs were disregarded.

## Interventions

Three cervical cancer screening strategies were compared. The descriptions of the interventions can be seen in Fig 1.

1. Cytology strategy (Brazilian standard screening program): conventional cytology to women aged 25 to 64 every three years;

2. hrHPV strategy (new ongoing screening program): hrHPV test to women aged 25 to 64 every five years, with triage of positive tests by liquid-based cytology;

3. Hybrid strategy (hypothetical): conventional cytology to women aged 25 to 29 years old every three years, and hrHPV test to women aged 30 to 64 every five years, with triage of positive tests by liquid-based cytology.

## Study design

Since cervical cancer is a long-term disease from HPV infection to the invasive lesion, the Markov model with microsimulation was chosen [23]. The model establishes health status, possible transitions, the probabilities of each status transition and values of costs and effects for each state. The TreeAge Pro 2018® software, Williamstown/USA, was used for economic modelling.

The model considered local data and international literature by age-groups. For population data and women's survival table, official data from São Paulo's statistical agency were used [24]. For the natural history of HPV infection, consequences of clinical management of precursor lesions, and cancer survival by stage, data from the literature review were used [25]. The probabilities of hrHPV infection were obtained from results of the first 30 months of the current program (data yet to be published). Data from the cytology-based old program were used for the prevalence of precursor lesions: aggregate data generated by the referral laboratory

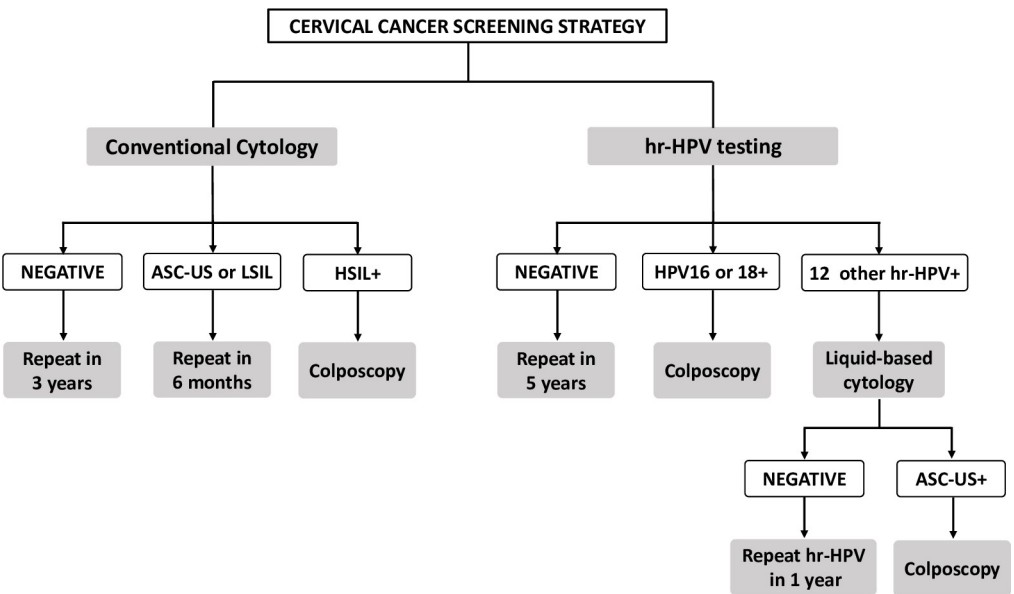

**Fig 1. Cervical cancer screening strategies: Conventional cytology (right) and primary hrHPV test (left).** The hybrid strategy is a mix of cytology for women aged 25 to 29 years and hrHPV test for women aged 30 to 64 years.

from the Women's Hospital at Unicamp. Data were obtained from recent data of long-term prospective studies for cytology and colposcopy/biopsy results after a hrHPV test [26]. The prevalence of cervical cancer stages in the region was obtained from local data [27]. The probability of adhering to the screening intervals and health services coverage to search for cases were also considered by the literature [28].

Some probabilities used by the model are in Table 1. In Fig 2 the prevalence of hrHPV and the prevalence of precursor lesions according to age are presented. The cumulative probabilities of dying from cervical cancer, based on the model, increase to 50 years old. After that, it becomes stable. It reaches the risk of six deaths per every thousand women without any intervention while screening by cytology or by hrHPV test decreases respectively to three or two deaths per every thousand women.

**Table 1. Probabilities used by the model to compare screening strategies in Indaiatuba, Brazil.**

| Variable | Probability |
|---|---|
| Primary health care coverage [28] | 0.39 |
| Satisfactory cytology [29] | 0.93 |
| Normal cytology (Unicamp, 2019) (local data) | 0.98 |
| Access to cytology result [30] | 0.76 |
| Abnormal Colposcopy [31] | 0.06 |
| Transition from CIN 1 to 2/3 [32] | 0.02 to 0.08 |
| Transition from CIN 2/3 to FIGO Stage I [32] | 0.10 |
| Transition from FIGO Stage I to II [32] | 0.23 |
| Transition from FIGO Stage II to III [32] | 0.33 |
| Transition from FIGO Stage III to IV [32] | 0.54 |

CIN: cervical intraepithelial neoplasia; FIGO: International Federation of Gynecology and Obstetrics.

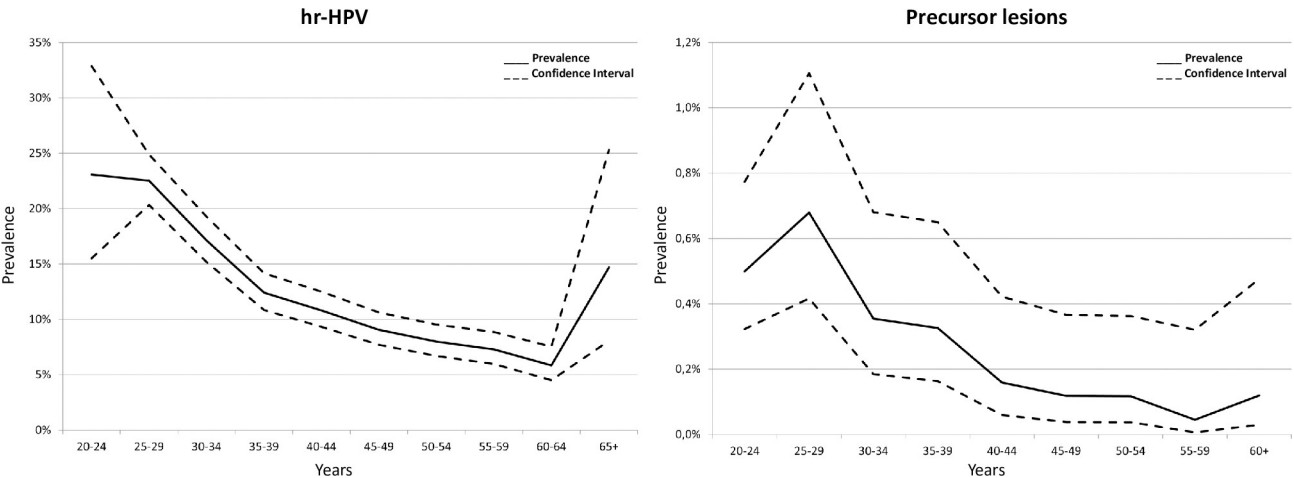

**Fig 2. Prevalence and 95% confidence intervals used for modelling of hrHPV and precursor lesions.**

A discount rate of 5% per year was used, with a sensitivity analysis of 0–10%, according to the Economic Evaluation Guideline of the Ministry of Health [33]. The results were compared with the Brazilian epidemiological models identified in the literature [34]. One million women were simulated in hypothetical cohorts after calibrating the model. The outcomes of cervical cancer were evaluated for the life expectancy of women.

The Ethics Committee of UNICAMP approved the study protocol (CAAE 43815315.9.0000.5404) and waived the Informed Consent Form's need.

## Effectiveness measures

According to imperfect states of health, the primary outcome assessed was quality-adjusted life-years (QALY), a two-dimensional measure of well-being, which adjusts years of life. The value is between zero (death) and one (perfect health). Negative values are also possible to state worse than death. QALY data by age were derived from the literature [35] and adjusted by cervical intraepithelial neoplasia (CIN) lesions or cervical cancer by the International Federation of Gynecology and Obstetrics (FIGO) [36].

## Estimated resources spent and costs

The resource items were based on the detailed clinical management description by the literature review and specialists consults. A mixed micro- and macro-costing technique obtained the measurement of resources. Micro-costing detailed the screening strategies by the observational method of all expenditures and collection of primary data. Macro-costing was based on administrative data from the Indaiatuba and the Women's Hospital of Unicamp, the cancer treatment referral centre.

The valuation of resources used triangulation of methods: direct measurement of costs at the primary health care facility where screening takes place, and at the laboratory of the Women's Hospital at the Unicamp, accounting method (costing by a cost) and standard unit costs (based on the Management Table System of SUS procedures and the Health Price Bank, available at the SUS Information Department) [37]. Costs were calculated in Reais (R$—Brazilian currency unit) and then converted to Dollars (US$) to facilitate interpretation and comparability (currency on July 1, 2020, US$ 1 = R$ 5,318). The main costs used are seen in Table 2.

Once the hrHPV test is not approved at the public health setting in Brazil, the cost was estimated by the top literature values that are being used in international locations (from US$ 10.00 to 30.00). The price at the private market in Brazil for one single test is around US$ 50.00.

## Sensitivity analysis

Analyzes of the adopted parameters' robustness were carried out, considering the variations in the probabilities, outcomes, and costs. Variations in the parameters were evaluated in univariate analyzes on a 'Tornado Diagram'. Probabilistic sensitivity analysis was also used to explore the uncertainties in terms of costs and effects on the model's parameters.

## Willingness to pay threshold

Brazil has no standard recommendation about the economic or health value on the willingness to pay threshold (WPT) but admits that economic valuations would accept the value of one to three times the per capita gross domestic product (pcGDP) per QALY [33]. In 2018 the Brazillian pcGDP was US$ 6,317.00 (R$ 33,593.82) [38].

# Results

Compared to conventional cytology, hrHPV testing with liquid-based cytology for triage of positive results and the hybrid strategy were dominant strategies for cervical cancer screening. Costs were lower and provided a more effective option at an economy of US$ 37.87 for the hybrid strategy and an economy of US$ 6.16 for the HPV strategy, per QALY gained (Table 3). The values indicate the hypothetical value of how much it would have to be spent per women to add one more year in good health with the strategies proposed.

## Sensitivity analysis

The univariate sensitivity analysis was performed to assess the stability of the model's assumptions on the study findings (Fig 3). Reduction in treatment costs would influence a reduction in ICER in both analyses. An increase in the costs of the hrHPV test would increase ICER. An

**Table 2. Valuation of costs used by the model to compare screening strategies in Indaiatuba/SP, Brazil.**

| Variable | Cost (US$) |
|---|---|
| Biopsy[a] | US$ 20.73 |
| Conventional cytology[a] | US$ 13.16 |
| Liquid-based cytology[a] | US$ 26.32 |
| HPV testing[b] | US$ 30.00 |
| Colposcopy[a] | US$ 7.63 |
| Outpatient appointment in primary health care[a] | US$ 4.37 |
| Cervical cancer FIGO stage I treatment[c] | US$ 599/5 years |
| Cervical cancer FIGO stage II treatment[c] | US$ 1,517/5 years |
| Cervical cancer FIGO stage III treatment[c] | US$ 2,273/5 years |
| Cervical cancer FIGO stage IV treatment[c] | US$ 2,423/5 years |

[a]Total expenditures involved in the procedure;

[b]Once HPV testing is not standard in Brazil, we used an estimated price for the public health setting;

[c]Total expenditures for treatment in five years at the referral free of charge hospital (Unicamp).

**Table 3. Incremental cost-effectiveness ratio results of modelling cervical cancer screening strategies in a population-based program in Indaiatuba, Brazil.**

| Strategy | Cost | QALY | ICER |
|---|---|---|---|
| Cytology (25-64y) | US$ 181.64 | 386.98 | — |
| Cytology (25-29y) + hrHPV (30-64y) | US$ 89.49 | 389.41 | (—) US$ 37.87 |
| hrHPV (25-64y) | US$ 158.12 | 390.80 | (—) US$ 6.16 |

QALY: quality-adjusted life-years; ICER: incremental cost-effectiveness ratio; (—): negative value.

increase in primary health care access coverage to the tests would discretely affect the hybrid strategy.

## Discussion

### Main findings

According to this modelling analyze that has incorporated setting-specific information into the risks and costs, the switch from cytology to hrHPV testing in the cervical cancer screening program of Indaiatuba is cost-effective and less costly [39]: the amount to spend to obtain one year with quality of life was far below the willingness to pay threshold (negative ICER of US$ 37.87 in the cytology plus hrHPV strategy and of US$ 6.16 in the hrHPV strategy, for a threshold of US$ 6,317.00). The variables that most influenced the ICER were the cost of the hrHPV test and the treatment's total costs.

### Strengths and limitations

There are two main strengths of this study. First, it used for modelling local screening data of past cytology-based and the current hrHPV testing screening program. Secondly, the program implemented is population-based, not a pilot study, as observed in the literature. We ignore another population-based modelling study coming from a low or middle-income country.

Models in economic analysis are based on many assumptions, some of which may be more accurate than others. They simplify real-life and fix performance parameters such as coverage, excess tests and others. The main limitation of this study is the inadequate precision of the cost of the hrHPV test. However, the sensitivity analysis has shown that the cost-effectiveness would happen under various scenarios.

### Interpretation

The hrHPV test cost is addressed as the main limitation for the implementation of a hrHPV-based screening program. In our model, representing an urban city from a middle-income country, the hrHPV testing strategy was more cost-effective considering the international literature's top price. However, it is essential to note that the model works in a fixed perspective, where tests are performed regularly. An opportunistic scenario expects lower intervals (excess tests), which would probably reduce the strategy's cost-effectiveness. It is indispensable that a shift to a hrHPV-based screening strategy would be followed by plans to improve the program's performance.

Other modelling studies have shown that hrHPV test screening is more cost-effective than cytology [9–16]. The reduction in screening costs associated with the lower cancer incidence and mortality when more precursor lesions are detected and treated is the main argument for noting positive results on cost-effectiveness when switching from cytology to hrHPV test, regardless of the attendance on screening [11, 40, 41]. The increase significantly influences the

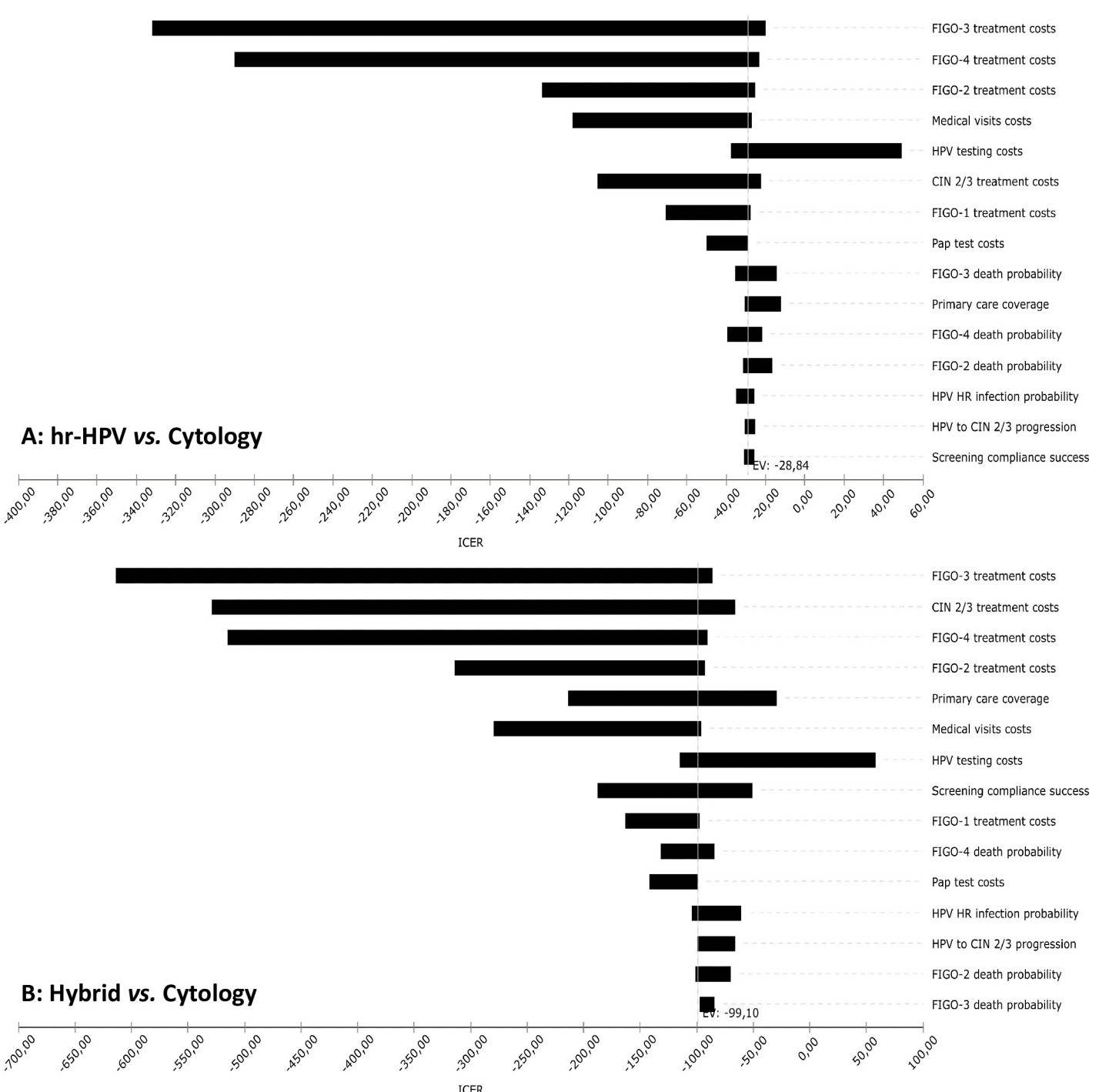

**Fig 3.** Tornado diagram of the sensitivity analysis for the hrHPV (a) and hybrid (b) versus cytology strategies models. *The diagram shows the range of the Incremental Cost-Effectiveness Ratio (ICER).*

cost-savings in the screening interval. The results are positive even when varying the settings of the studies. Few reports have incorporated local settings data. Jansen *et al.* reported similar results after the first years of hrHPV screening implementation in the Dutch program [11].

The cost of the treatment was another variable that influenced the result. In Brazil, treatment is free of care, so a reduction in incidence and advanced cases may significantly impact public health. Almost three in every four cervical cancer cases in Brazil are diagnosed in Stages II or more advanced [42]. In this study, it was assumed that all patients were diagnosed and received appropriate standard treatment. It may have overestimated the long term effects of screening.

The influence on the results of coverage on screening was discrete in the sensitivity analysis. High coverage should be the main target of any screening program, organized or opportunistic. However, in modelling studies, the inputs consider a fixed assurance of screening components: invitation, testing, and further assessment. Quality assurance was not accessed in this study. An economic evaluation post-implementation is expected to address these issue.

Another critical query in screening programs is the superiority of hrHPV test accuracy when compared to cytology accuracy. It is widely known that the conventional cytology's sensitivity is under the threshold desirable, particularly in LMIC [5]. In places where the quality control of cytology screening is fragile, the WHO suggests that a switch to hrHPV testing or even visual inspection with acetic acid (VIA) should be considered [8]. The automation in processing samples in hrHPV test is expected to improve the sensitivity and the accuracy of the screening test.

However, in our modelling study, the cytology's accuracy–conventional or liquid-based did not influence the results. It might be explained by the high impact expected of cytology in screening, even when sensitivity is moderate to low. Historical evidence pointed to a cervical cancer risk reduction of 46% with a single life-time cytology test [43]. Even considering that cervical cancer incidence rates are still moderate to high in low and middle-income countries, a significant reduction was observed in countries with established opportunistic cytology-based screening programs [1]. The recommendation to consider switching to hrHPV testing should be evaluated under this perspective.

The hrHPV test performance is improved when women are over 30 years old, bypassing its low specificity in young women. In Brazil, target women for screening are those from 25 to 64 years old. When implementing hrHPV testing in screening, some would not be comfortable testing women under 30 years old. The policymakers of Indaiatuba choose to use a widely licensed hrHPV testing with genotyping HPV16 and 18 to increasing specificity in youngers and apply to all target women, arguing that keeping both logistics (cytology and hrHPV test) would be confusing and costly. This study showed that the decision might be correct by the economic view: the hrHPV testing strategy would be as cost-effective as the hybrid strategy. The results of the economic evaluation post-implementation probably will support this decision.

## Conclusions

Based on real-life data, the decision to switch to hrHPV testing in Indaitauba was cost-effective. This result may support Brazil and other middle-income countries policymakers to similar practices. The best implementation strategy is yet to be defined, but we believe that the single hrHPV strategy should be recommended, avoiding logistics challenges regarding a hybrid strategy. Still, the results showed in this modelling study are in line with the literature. There is a clear tendency to support hrHPV testing on screening, even when coverage is not as high as observed in organized screening programs.

## Author Contributions

**Conceptualization:** Diama Bhadra Vale, Julio Cesar Teixeira, Luiz Carlos Zeferino.

**Data curation:** Marcus Tolentino Silva, Michelle Garcia Discacciati.

**Formal analysis:** Diama Bhadra Vale, Marcus Tolentino Silva, Julio Cesar Teixeira.

**Funding acquisition:** Julio Cesar Teixeira.

**Investigation:** Diama Bhadra Vale, Ilana Polegatto, Julio Cesar Teixeira.

**Methodology:** Diama Bhadra Vale.

**Project administration:** Michelle Garcia Discacciati, Julio Cesar Teixeira.

**Supervision:** Julio Cesar Teixeira, Luiz Carlos Zeferino.

**Validation:** Diama Bhadra Vale.

**Writing – original draft:** Diama Bhadra Vale, Julio Cesar Teixeira.

**Writing – review & editing:** Diama Bhadra Vale, Marcus Tolentino Silva, Michelle Garcia Discacciati, Ilana Polegatto, Julio Cesar Teixeira, Luiz Carlos Zeferino.

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
