## [Decision Letter · Decision Letter 0]

19 Mar 2021

PONE-D-21-05775

Is the HPV-test more cost-effective than cytology in cervical cancer screening? An economic analysis from a middle-income country

PLOS ONE

Dear Dr. Vale,

Thank you for submitting your manuscript to PLOS ONE. After careful consideration, we feel that it has merit but does not fully meet PLOS ONE’s publication criteria as it currently stands. Therefore, we invite you to submit a revised version of the manuscript that addresses the points raised during the review process.

We look forward to receiving your revised manuscript.

Kind regards,

Luca Giannella

Academic Editor

PLOS ONE

Journal Requirements:

"This project is funded by UNICAMP (Women's Hospital), Indaiatuba City (SUS), and Roche Diagnostics, as detailed below: This study proposal (screening program implementation and cost-effectiveness analyses) was designed by researchers from UNICAMP and introduced by the Indaiatuba City Hall. Both UNICAMP and the municipality use the existing and functioning structure to place the new screening program and carry out the proposed study at no additional cost. The supplies and equipment required to perform HPV testing (for one round in five years), computer system development support, two lab technicians, and a screening program coordinator are provided or supported by Roche Diagnostics. Roche Diagnostics supports an external statistician defined by researchers to develop a model and perform the planned cost-effectiveness analyses. There are no planned compensations or cash transfers provided to any institution or researchers stated in the parties' cooperation agreement."

We note that you received funding from a commercial source: Roche Diagnostics.

Reviewers' comments:

Reviewer's Responses to Questions

**Comments to the Author**

1. Is the manuscript technically sound, and do the data support the conclusions?

Reviewer #1: Yes

Reviewer #2: Partly

Reviewer #3: Yes

2. Has the statistical analysis been performed appropriately and rigorously? 

Reviewer #1: I Don't Know

Reviewer #2: I Don't Know

Reviewer #3: Yes

3. Have the authors made all data underlying the findings in their manuscript fully available?

Reviewer #1: No

Reviewer #2: No

Reviewer #3: Yes

4. Is the manuscript presented in an intelligible fashion and written in standard English?

Reviewer #1: Yes

Reviewer #2: Yes

Reviewer #3: Yes

5. Review Comments to the Author

Reviewer #1: Thank you for the opportunity to read this paper. This issue is of great interest for implementation in a country like Brazil. The cost-effectiveness analysis of screening strategies of the most prevalent gynecological cancer in Brazil is extremely relevant.

I would like to express some comments and make some questions. Their are small corrections and suggestions for the text.

1) The test used includes the analysis of which hrHPV? Only 16/18? In the different states of Brazil, we know that the prevalence of HPV types is different and, depending on the analysis, it could eventually impair the analysis.

2) Line 93: I believe that you meant from 30 years old and not until 30 years old.

3) The sentence: "HPV vaccination in Brazil started in 2014, and the first cohort of vaccinated girls will not reach the age of screening until 2026." is out of context in my point of view.

4) The paragraph of line 100 could be omitted and references should only be cited when necessary.

5) In several parts of the text, the FIGO classification is written in Arabic numbers and I believe that Roman numbers would be better even to avoid confusing the reading of CIN 2/3.

6) In line 160 there is an extra comma.

7) In the paragraph of line 166, you mentioned that QUALY is a two-dimensional measure and that it varies between zero and one. Based on this afirmation, could you better explain the data in Table 3, please?

8) Line 229, it was not clear which was the value of willingness to pay for the new proposition.

9) In your opinion, the hrHPV test is the best strategy for Brazil?

I wait for your considerations. Thank you so much.

Reviewer #2: Dear authors,

I think you need to clarify deeply the clinically validated HPV assays that you used. Moreover, a clear and sound explanation with results/facts from the population studied about the effectiveness/detection of precursor lesions/ difference in morbidity and mortality in support of HPV testing.

Data should be made available.

The cost-effectiveness should be made obvious.

Reviewer #3: In this article, the authors discussed screening methods for precancerous lesions of cervical cancer. In particular, They were considering cost-effectiveness in low-income and middle-income countries.

I think it is very informative and interesting. It is well accepted without any particular indication. I think it is suitable and acceptable for this journal.

I don't have much to point out, so I don't feel the need for correction. The strengths and limitations are also well considered.

It's close to what has already been reported, but I think it's a great treatise that reflects the realities of the region rather than novelty.

6. PLOS authors have the option to publish the peer review history of their article (what does this mean?). If published, this will include your full peer review and any attached files.

Reviewer #1: No

Reviewer #2: No

Reviewer #3: No

---

## [Author Response · Author response to Decision Letter 0]

21 Apr 2021

Response to the Editor

Dear editor,

I have made some changes marked at cover letter upon your request.

Best regards,

DBV.

Response to Reviewers

Reviewer #1: 

1. The test used includes the analysis of which hrHPV? Only 16/18? In the different states of Brazil, we know that the prevalence of HPV types is different and, depending on the analysis, it could eventually impair the analysis.

R: The test used is the Cobas® HPV Test (Roche Molecular Systems, Pleasanton, CA) that provides HPV 16 and HPV 18 genotyping and pooled 12-other hrHPV (types 31, 33, 35, 39, 45, 51, 52, 56, 58, 59, 66 and 68). HPV16 and 18 predominate in all places in the world, including in different Brazilian regions. After 30 months of the Indaiatuba program, preliminary results exhibited 173 HSIL/AIS detected with 59% related to HPV16 and/or 18 (expected 50%). Among 17 cervical cancer detected, 76% were related to HPV16 and/or 18 (expected 70%). The performance of the chosen test was higher than expected, and we believe that the possible low differences in the HPV type distribution by region are not a relevant concern.

We added some information about the program and the HPV test considered:

Lines 85-86: In 2017 Indaiatuba adopted the hrHPV test to replace cytology in cervical cancer screening. A detailed description of the program was previously published by Teixeira and colleagues [18]. In summary, the hrHPV test chosen provides individual results on the highest risk genotypes - HPV 16 and HPV 18 - and an aggregated results on the twelve other hr-HPV genotypes (types 31, 33, 35, 39, 45, 51, 52, 56, 58, 59, 66 and 68). The negative test indicates a repetition in five years.

Lines 300-302: The policymakers of Indaiatuba choose to use a widely licensed hrHPV testing with genotyping HPV16 and 18 to increasing specificity in youngers and apply to all target women, arguing that keeping both logistics (cytology and hrHPV test) would be confusing and costly.

2. Line 93: I believe that you meant from 30 years old and not until 30 years old.

R: Corrected in the text.

3. The sentence: "HPV vaccination in Brazil started in 2014, and the first cohort of vaccinated girls will not reach the age of screening until 2026." is out of context in my point of view.

R: Removed from the text.

4. The paragraph of line 100 could be omitted and references should only be cited when necessary.

R: Removed from the text.

5. In several parts of the text, the FIGO classification is written in Arabic numbers and I believe that Roman numbers would be better even to avoid confusing the reading of CIN 2/3.

R: Corrected in the text.

6. In line 160 there is an extra comma.

R: Removed from the text.

7. In the paragraph of line 166, you mentioned that QUALY is a two-dimensional measure and that it varies between zero and one. Based on this afirmation, could you better explain the data in Table 3, please?

R: Added the following sentence in Results: “The values indicate the hypothetical value of how much it would have to be spent per women to add one more year in good health with the strategies proposed.” 

8. Line 229, it was not clear which was the value of willingness to pay for the new proposition.

R: Added the following sentence: “…negative ICER of US$ 37.87 in the cytology plus hrHPV strategy and of US$ 6.16 in the hrHPV strategy, for a thereshold of US$ 6,317.00. “ 

9. In your opinion, the hrHPV test is the best strategy for Brazil?

R: Added the following sentence in the conclusion: “…we believe that the single hrHPV strategy should be recommended, avoiding logistics challenges regarding a hybrid strategy.”

Reviewer #2:

1. I think you need to clarify deeply the clinically validated HPV assays that you used. Moreover, a clear and sound explanation with results/facts from the population studied about the effectiveness/detection of precursor lesions/ difference in morbidity and mortality in support of HPV testing.

R: Dear reviewer. We appreciate and agree that the clinical results are extremely important, and we are drafting another publication in this issue. The final analysis is still not ready. However, this is not the objective of this study. This paper aims to report the economic analysis of cost-effectiveness by the public health system's perspective. This is the research question at the moment in Brazil regarding cervical cancer; that is why we are focusing on the publication of these results. 

2. Data should be made available.

R: We appreciate your recommendation and created a repository with the files: www.doi.org/10.17605/osf.io/f3ygd

3. The cost-effectiveness should be made obvious.

R: This feedback is essential, and it is in line with some of the comments of “Reviewer #1”. We added some sentences expecting to improve the writing to make this question clear: 1) R: Added the following sentence in Results: “The values indicate the hypothetical value of how much it would have to be spent per women to add one more year in good health with the strategies proposed.”; 2) R: Added the following sentence: “… negative ICER of US$ 37.87 in the cytology plus hrHPV strategy and of US$ 6.16 in the hrHPV strategy, for a thereshold of US$ 6,317.00.”; 3) R: Added the following sentence in the conclusion: “…we believe that the single hrHPV strategy should be recommended, avoiding logistics challenges regarding a hybrid strategy.”

Reviewer #3:

I think it is very informative and interesting. It is well accepted without any particular indication. I think it is suitable and acceptable for this journal.

I don't have much to point out, so I don't feel the need for correction. The strengths and limitations are also well considered.

R: We really appreciate your comments and sincerely expect that our results will be informative to support decisions regarding screening in Brazil and other middle-income countries.

---

## [Editor Report · Decision Letter 1]

3 May 2021

Is the HPV-test more cost-effective than cytology in cervical cancer screening? An economic analysis from a middle-income country

PONE-D-21-05775R1

Dear Dr. Vale,

We’re pleased to inform you that your manuscript has been judged scientifically suitable for publication and will be formally accepted for publication once it meets all outstanding technical requirements.

Kind regards,

Luca Giannella

Academic Editor

PLOS ONE
---

## [Editor Report · Acceptance letter]

6 May 2021

PONE-D-21-05775R1 

Is the HPV-test more cost-effective than cytology in cervical cancer screening? An economic analysis from a middle-income country 

Dear Dr. Vale:

I'm pleased to inform you that your manuscript has been deemed suitable for publication in PLOS ONE. Congratulations! Your manuscript is now with our production department. 

Kind regards, 

on behalf of

Dr. Luca Giannella 

Academic Editor

PLOS ONE